# ONLINE CHANGE POINT DETECTION FOR MULTIVARI-ATE POISSON POINT PROCESSES

## ABSTRACT

We study online change point detection for multivariate inhomogeneous Poisson point process data streams. Although this setting is common in applications such as earthquake seismology, climate monitoring, and epidemic surveillance, it remains largely underexplored in the statistics and data science literature. We propose a method that uses low-rank matrices to represent the multivariate Poisson intensity function, resulting in an adaptive procedure to detect local changes in a nonparametric setting. Our algorithm processes the stream in a single-pass, and the per-observation cost is a constant independent of the elapsed stream length. We also provide theoretical guarantees to control the overall false alarm probability and quantify the detection delay. Numerical experiments demonstrate that our method is statistically robust and computationally efficient.

## 1 INTRODUCTION

An inhomogeneous Poisson point process models random events that occur independently in space at location–dependent rates. Examples include forest fires (Stoyan & Penttinen, 2000), earthquakes (Bray & Schoenberg, 2013), citywide crime incidents (Baddeley et al., 2021), and epidemic outbreaks (Al-Dousari et al., 2021).

The intensity function is central to understanding point processes, as it specifies the expected number of events per unit area at any given location. More precisely, a Poisson point process sample $X \subset \mathbb{R}^d$ is said to sampled from an intensity $\lambda : \mathbb{R}^d \to \mathbb{R}^+$ if:

**(1)** for every set $S \subseteq \mathbb{R}^d$, the count $|S \cap X|$ is a Poisson random variable with mean $\int_S \lambda(x)\,dx$; and

**(2)** for disjoint sets $S_1, \dots, S_n \subset \mathbb{R}^d$, the random variables $|S_1 \cap X|, \dots, |S_n \cap X|$ are independent.

Estimation for a *single* Poisson point process is well studied: Reynaud-Bouret (2003) derive minimax rates of estimating the intensity function in one dimension; Flaxman et al. (2017) estimate intensities using RKHS; and Xu et al. (2025) use tensor techniques for intensity estimation.

In this paper, we focus on detecting changes in a *stream* of Poisson point processes in an online setting. The streaming data follows one of the two scenarios. In the first, the intensity does not change, and

$$\{X^{(i)}\}_{i=1}^\infty \overset{i.i.d.}{\sim} \lambda^*.$$

In the second, the intensity changes at an unknown change point $\mathfrak{b}$, and

$$\{X^{(i)}\}_{i=1}^{\mathfrak{b}} \overset{i.i.d.}{\sim} \lambda^* \quad \text{and} \quad \{X^{(i)}\}_{i=\mathfrak{b}+1}^\infty \overset{i.i.d.}{\sim} \lambda_a^*.$$

Online change point detection for multivariate Poisson point process streams has wide-ranging applications, including earthquake seismology (Ogata, 2011), wildfire monitoring (Xu & Schoenberg, 2011), and epidemic surveillance (Hohl et al., 2020). However, to our knowledge, there is no method that reliably detects changes in the intensity function in the change point literature.

### 1.1 RELATED WORK

Online nonparametric change point detection has been studied extensively. Among many contributions, Li et al. (2015) developed a computationally efficient $M$-statistic for kernel-based methods,

Shin et al. (2022) introduced the *e*-detector for simultaneous parametric and nonparametric detection, and Romano et al. (2023) proposed Functional Online CuSUM with constant per-iteration cost. However, most existing work targets changes in multivariate *densities*. It is unclear whether their computational efficiency and theoretical guarantees extend to Poisson point process streams.

## 1.2 SUMMARY OF RESULTS

**New algorithm for online change detection.** We introduce a new online procedure for detecting changes in Poisson point processes that is highly efficient. In particular, the per-observation cost is constant independent of the elapsed stream length. As a result, our method is single-pass, and the total computational cost is linear in the stream length.

**General theory.** We develop a theoretical framework for online change detection in multidimensional Poisson point processes. In our framework, each realization $X^{(i)}$ is represented by a low-rank intensity matrix, yielding a scalable algorithm adaptive to local changes in the intensity. The framework controls both approximation bias and stochastic variance in the Poisson point process streaming data settings.

**Finite-sample guarantees.** We establish nonasymptotic guarantees to show that our newly proposed method both controls the false alarm probability and can detect a true change within a delay that depends explicitly on the jump size $\|\lambda^* - \lambda_a^*\|_{\mathbf{L}_2}$.

**Empirical evidence.** We demonstrate that our procedure reliably identifies meaningful intensity changes while remaining computationally efficient through extensive simulations and a real-data application.

## 1.3 NOTATIONS

Let $\mathcal{M} \in \mathbb{R}^{m \times n}$ be a generic matrix. For $\mu \in \{1, \ldots, m\}$ and $\eta \in \{1, \ldots, n\}$, let $\mathcal{M}_{(\mu, \eta)}$ denote the $(\mu, \eta)$ entry of $\mathcal{M}$. The Frobenius norm satisfies $\|\mathcal{M}\|_{\mathrm{F}}^2 = \sum_{\mu=1}^m \sum_{\eta=1}^n \mathcal{M}_{(\mu, \eta)}^2$. Let $\mathrm{Rank}(\mathcal{M})$ denote the rank of $\mathcal{M}$. Suppose $\mathrm{Rank}(\mathcal{M}) = s$, and let $\{\sigma_k(\mathcal{M})\}_{k=1}^s$ denote the singular values of $\mathcal{M}$. The operator norm of $\mathcal{M}$, denoted by $\|\mathcal{M}\|$, is the largest singular value of $\mathcal{M}$.

Let the singular value decomposition (SVD) of $\mathcal{M}$ satisfy $\mathcal{M} = U\Sigma V^\top$. Here $U \in \mathbb{R}^{m \times s}$ and $V \in \mathbb{R}^{n \times s}$ have orthonormal columns, and $\Sigma = \mathrm{diag}\left(\sigma_1(\mathcal{M}), \ldots, \sigma_s(\mathcal{M})\right) \in \mathbb{R}^{s \times s}$. For $r \leq s$, denote the truncated SVD of $\mathcal{M}$ by

$$\mathrm{SVD}(\mathcal{M}, r) = U\,\Sigma_{(r)}\,V^\top, \qquad \Sigma_{(r)} = \mathrm{diag}\left(\sigma_1(\mathcal{M}), \ldots, \sigma_r(\mathcal{M}), 0, \ldots, 0\right) \in \mathbb{R}^{s \times s}.$$

For a domain $\Omega \subset \mathbb{R}^d$, define $\mathbf{L}_2(\Omega) = \left\{ f : \Omega \to \mathbb{R} \ : \ \|f\|_{\mathbf{L}_2}^2 = \int_\Omega f^2(x)\, dx < \infty \right\}$. Here $dx$ indicates that the integration is with respect to the Lebesgue measure. We say $\{\phi_k\}_{k=1}^\infty$ is an orthonormal family in $\mathbf{L}_2(\mathbb{R})$ if

$$\int_{\mathbb{R}} \phi_k(x)\, \phi_j(x)\, dx = \begin{cases} 1, & \text{if } k = j; \\ 0, & \text{if } k \neq j. \end{cases}$$

For $\beta \geq 1$, let $W^{2,\beta}(\mathbb{R}^d)$ denote the Sobolev space

$$W^{2,\beta}(\mathbb{R}^d) = \left\{ f \in \mathbf{L}_2(\mathbb{R}^d) \ : \ f^{(b)} \in \mathbf{L}_2(\mathbb{R}^d) \text{ for all multi-indices } b = (b_1, \ldots, b_d) \text{ with } |b| \leq \beta \right\},$$

where $|b| = b_1 + \cdots + b_d$ and $f^{(b)} = \frac{\partial^{|b|} f}{\partial x_1^{b_1} \cdots \partial x_d^{b_d}}$. The corresponding Sobolev norm of $f$ is $\|f\|_{W^{2,\beta}}^2 = \sum_{b:|b| \leq \beta} \|f^{(b)}\|_{\mathbf{L}_2}^2$.

## 2 ONLINE CHANGE POINT DETECTION FOR POISSON POINT PROCESSES

We begin with a detailed description of our streaming model. In the training phase, we observe a collection of Poisson processes $X^{(i)} \subset \mathbb{R}^d$ with $i = 1, \ldots, N_{\text{train}}$ independently sampled from a common intensity $\lambda^* : \mathbb{R}^d \to \mathbb{R}^+$. We focus on the multivariate settings with $d \geq 2$. The case $d = 1$ is discussed in Remark 3.

After the training phase, the stream follows one of two scenarios. In the first, the intensity of the Poisson point processes does not change: $\{X^{(i)}\}_{i=1}^{\infty} \stackrel{i.i.d.}{\sim} \lambda^*$. In the second, the intensity changes at an unknown change point $\mathfrak{b} \geq N_{\text{train}}$: $\{X^{(i)}\}_{i=1}^{\mathfrak{b}} \stackrel{i.i.d.}{\sim} \lambda^*$ and $\{X^{(i)}\}_{i=\mathfrak{b}+1}^{\infty} \stackrel{i.i.d.}{\sim} \lambda_a^*$.

Our goal is to develop an online algorithm such that (i) when there is no change, the overall probability of a false alarm is kept small; and (ii) if a change occurs after the training phase, the algorithm raises an alarm as quickly as possible to minimize the detection delay.

To this end, we take a different approach from the existing literature. We map continuous intensity functions to matrices with a distance-preserving property. This enables efficient operation over arbitrary intervals via dynamic programming and avoids computing $\mathbf{L}_2$ norms of functions via additional Monte Carlo procedures in higher dimensions.

We begin by introducing the necessary notations. Let $x = (x_1, \ldots, x_d) \in \mathbb{R}^d$, and consider the coordinate split $(y, z) \in \mathbb{R}^{p+q}$,

$$y = (x_1, \ldots, x_p), \quad z = (x_{p+1}, \ldots, x_{p+q}), \quad p + q = d. \tag{1}$$

Let $\{\phi_k\}_{k=1}^{\infty}$ form orthonormal univariate basis of $\mathbf{L}_2(\mathbb{R})$. Then $\left\{\phi_{i_1}(x_1) \cdots \phi_{i_p}(x_p)\right\}_{i_1,\ldots,i_p=1}^{\infty}$ is a set of complete basis functions of $\mathbf{L}_2(\mathbb{R}^p)$. For any positive integer $M$, the collection of functions $\left\{\phi_{i_1}(x_1) \cdots \phi_{i_p}(x_p)\right\}_{i_1,\ldots,i_p=1}^{M}$ is orthonormal in $\mathbb{R}^p$ with cardinality being $M^p$. For convenience, by ordering the multi-indices $(i_1, \ldots, i_p)$ and $(\ell_1, \ldots, \ell_q)$ if necessary, we denote

$$\left\{\Phi_\mu(y)\right\}_{\mu=1}^{M^p} = \left\{\phi_{i_1}(x_1) \cdots \phi_{i_p}(x_p)\right\}_{i_1,\ldots,i_p=1}^{M} \subset \mathbb{R}^p,$$
$$\left\{\Psi_\eta(z)\right\}_{\eta=1}^{M^q} = \left\{\phi_{\ell_1}(x_{p+1}) \cdots \phi_{\ell_q}(x_{p+q})\right\}_{\ell_1,\ldots,\ell_q=1}^{M} \subset \mathbb{R}^q. \tag{2}$$

Let $\lambda^* : \mathbb{R}^{p+q} \to \mathbb{R}^+$ satisfy $\|\lambda^*\|_{W^{\beta,2}} < \infty$. Define the matrix $\mathcal{M}(\lambda^*) \in \mathbb{R}^{M^p \times M^q}$ by

$$\mathcal{M}(\lambda^*)_{(\mu,\eta)} = \iint_{\mathbb{R}^{p+q}} \lambda^*(y, z)\, \Phi_\mu(y)\, \Psi_\eta(z)\, dy\, dz. \tag{3}$$

Using $\mathcal{M}(\lambda^*)$, we can approximate $\lambda^*$ by its truncated expansion:

$$\lambda_M^*(y, z) = \sum_{\mu=1}^{M^p} \sum_{\eta=1}^{M^q} \mathcal{M}(\lambda^*)_{\mu,\eta}\, \Phi_\mu(y)\, \Psi_\eta(z). \tag{4}$$

It was shown in Appendix G.1 of Peng et al. (2024) that if $\{\phi_k\}_{k=1}^{\infty}$ are univariate Legendre polynomials, then

$$\|\lambda^* - \lambda_M^*\|_{\mathbf{L}_2} \leq C\, \|\lambda^*\|_{W^{\beta,2}}\, M^{-\beta}. \tag{5}$$

**Remark 1** (Coordinate split). *It follows from* (4) *and* (5) *that* $\mathcal{M}(\lambda^*)$ *provides an accurate matrix representation of* $\lambda^*$ *with small approximation error. This representation requires a coordinate split in* $\mathbb{R}^d$. *As suggested by Theorem* 2 *in the Appendix, such a split is valid for any function that admits a functional PCA representation. The split can also be specified using prior knowledge of the dataset, as demonstrated in our real-data example. As a third option, one can partition the features into two groups so that variables are more correlated within groups and less correlated across groups.*

For each process $X^{(i)}$, define its intensity matrix by

$$\widehat{\mathcal{M}}^{(i)} \in \mathbb{R}^{M^p \times M^q}, \qquad \widehat{\mathcal{M}}^{(i)}_{(\mu,\eta)} = \sum_{x=(y,z)\in X^{(i)}} \Phi_\mu(y)\, \Psi_\eta(z). \tag{6}$$

Suppose $\{X^{(i)}\}_{i=1}^{\mathfrak{b}} \stackrel{i.i.d.}{\sim} \lambda^*$. It follows from Lemma 8 that $\mathbb{E}(\widehat{\mathcal{M}}^{(i)}_{(\mu,\eta)}) = \mathcal{M}(\lambda^*)_{(\mu,\eta)}$. Thus if the intensity function changes at $\mathfrak{b}$, we have that

$$\mathbb{E}(\widehat{\mathcal{M}}^{(i)}) = \begin{cases} \mathcal{M}(\lambda^*) & \text{if } i \leq \mathfrak{b}; \\ \mathcal{M}(\lambda_a^*) & \text{if } i > \mathfrak{b}. \end{cases} \tag{7}$$

Consequently, for any $n \leq \mathfrak{b}$, the deviation between the matrices $\frac{1}{n}\sum_{i=1}^{n}\widehat{\mathcal{M}}^{(i)}$ and $\mathcal{M}(\lambda^*)$ can be controlled by standard large-sample bounds.

Our online change detection procedure is summarized in Algorithm 1. Below we briefly explain its implementation. Using dynamic programming, at time $t = j$ and for any $k \in \{1, \ldots, W\}$, we maintain

$$L[k] = \sum_{i=1}^{j-W-1+k} \widehat{\mathcal{M}}^{(i)} \quad \text{and} \quad R[k] = \sum_{i=j-W+k}^{j} \widehat{\mathcal{M}}^{(i)}.$$

Hence, for a given pair $(j, k)$, the matrix $D \in \mathbb{R}^{M^p \times M^q}$ in (11) of Algorithm 1 is the CUSUM statistic

$$D = \frac{1}{j-W-1+k} \sum_{i=1}^{j-W-1+k} \widehat{\mathcal{M}}^{(i)} - \frac{1}{W-k+1} \sum_{i=j-W+k}^{j} \widehat{\mathcal{M}}^{(i)}, \tag{8}$$

which compares the data between the intervals $[1, j-W-1+k]$ and $[j-W+k, j]$. For example if $j = \mathfrak{b} + W$ and $k = 1$, from (7) we can deduce that

$$D = \frac{1}{\mathfrak{b}}\sum_{i=1}^{\mathfrak{b}} \widehat{\mathcal{M}}^{(i)} - \frac{1}{W}\sum_{i=\mathfrak{b}+1}^{\mathfrak{b}+W} \widehat{\mathcal{M}}^{(i)} \approx \mathcal{M}(\lambda^*) - \mathcal{M}(\lambda_a^*) = \mathcal{M}(\lambda^* - \lambda_a^*), \tag{9}$$

where the last equality uses the linearity of the matrix representation. To further reduce variance when estimating $\lambda^* - \lambda_a^*$, we apply the restricted SVD procedure to $D$ as described in (12) of Algorithm 2.

Algorithm 2 has two components: (i) zeroing out higher-order entries of $D$ by trimming to an adaptive basis size, and (ii) applying SVD to the trimmed matrix. The trimming is adaptive to the sample size in (8): as $k$ ranges from 1 to $W$, the effective sample size is $W - k + 1$. The necessity of trimming comes from the fact that smaller samples only allow us to reliably estimate a smaller number of matrix coefficients.

We apply SVD to $D$ because its population counterpart $\lambda^* - \lambda_a^*$ is typically approximately low rank. Since $\lambda^* - \lambda_a^* \in \mathbf{L}_2(\mathbb{R}^{p+q})$, the functional singular value decomposition (Theorem 2) yields

$$\lambda^* - \lambda_a^* = \sum_{k=1}^{\infty} \sigma_k(\lambda^* - \lambda_a^*)\, f_k^*(y)\, g_k^*(z), \tag{10}$$

with nonincreasing singular values $\sigma_1(\lambda^* - \lambda_a^*) \geq \sigma_2(\lambda^* - \lambda_a^*) \geq \cdots \geq 0$ such that

$$\sum_{k=1}^{\infty} \sigma_k^2(\lambda^* - \lambda_a^*) = \|\lambda^* - \lambda_a^*\|_{\mathbf{L}_2}^2 < \infty,$$

and orthonormal functions $\{f_k^*(y)\} \subset \mathbf{L}_2(\mathbb{R}^p)$, $\{g_k^*(z)\} \subset \mathbf{L}_2(\mathbb{R}^q)$.

It is a commonly used assumption in the literature (e.g., Hall et al. (2006); Raskutti et al. (2012)) that, if $\|\lambda^* - \lambda_a^*\|_{W^{\beta,2}} < \infty$, then the singular values of $\lambda^* - \lambda_a^*$ decay at a polynomial or exponential rate. Since $\mathcal{M}(\lambda^* - \lambda_a^*)$ provides an accurate matrix representation of $\lambda^* - \lambda_a^*$, we can anticipate that the singular values of $\mathcal{M}(\lambda^* - \lambda_a^*)$, and consequently the singular values of $D$ in (9), decay at the same rate.

**Remark 2** (Computational cost). *Due to the dynamic programming design, for a new observation the computational cost of Algorithm 1 is $O\big(rW^{\,1+d/(2\beta+p\vee q)}\big)$. More precisely, at time $j$, updating each matrix in the lists $L$ and $R$ costs $O\big(W^{\,d/(2\beta+p\vee q)}\big)$. Computing the rank-$r$ SVD for each difference matrix $D$ in (11) costs $O\big(rW^{\,d/(2\beta+p\vee q)}\big)$. Since $L$ and $R$ each contain $W$ matrices, the total cost is $O\big(rW^{\,1+d/(2\beta+p\vee q)}\big)$. Consequently, the method is single-pass over the data stream, and the cost per data does not grow with the elapsed stream length.*

**Remark 3** (Poisson point process streaming data in 1D). *Algorithm 1 tackles online change point detection for Poisson point process streaming data in $\mathbb{R}^d$ with $d \geq 2$. On the other hand, in Section D*

*of the Appendix, we present a simplified one-dimensional variant that handles Poisson point process streams in $\mathbb{R}$ by representing the intensity as a vector rather than a matrix. The univariate setting is substantially simpler than the multivariate setting, as univariate nonparametric models do not suffer from the curse of dimensionality.*

---

**Algorithm 1:** Online multivariate Poisson point processes change detection

---

**Input**: Smoothness parameter $\beta > 0$; dimensionality $p, q$ with $p + q = d$; rank $r$; window size $W$; threshold constant $\mathcal{C}_\alpha$

**Initialization Stage**;

$M = \lceil (W/r)^{1/(2\beta + p \vee q)} \rceil$

▶ **Build a list $L$ of size $W$;**

**for** $k \leftarrow 1$ **to** $W$ **do**

$\qquad L[k] \leftarrow \displaystyle\sum_{i=1}^{N_{\text{train}} - W + k - 1} \widehat{\mathcal{M}}^{(i)}$; here $\widehat{\mathcal{M}}^{(i)}$ is computed via (6) with

**Detection Stage**;

ALARM $\leftarrow$ FALSE;

**for** $j \leftarrow N_{\text{train}} + 1, N_{\text{train}} + 2, \ldots$ **do**

$\qquad$ ▶ **Update the list $L$;**

$\qquad$ **for** $k \leftarrow 1$ **to** $W - 1$ **do**

$\qquad\qquad L[k] \leftarrow L[k+1]$;

$\qquad L[W] \leftarrow L[W] + \widehat{\mathcal{M}}^{(j-1)}$;

$\qquad$ ▶ **Form the list $R$ of size $W$;**

$\qquad$ **for** $k \leftarrow 1$ **to** $W$ **do**

$\qquad\qquad R[k] \leftarrow \displaystyle\sum_{i=j-W+k}^{j} \widehat{\mathcal{M}}^{(i)}$;

$\qquad$ ▶ **Compute the CUSUM statistics in the sliding window;**

$\qquad$ **for** $k \leftarrow 1$ **to** $W$ **do**

$$D \leftarrow \frac{L[k]}{n_1} - \frac{R[k]}{n_2} \quad \text{where} \quad n_1 \leftarrow j - W - 1 + k, \quad n_2 \leftarrow W - k + 1 \quad (11)$$

$$\text{Val} \leftarrow \text{Restricted SVD}\big(D, r, n_2, p, q, \beta\big) \quad \text{(see Algorithm 2)} \quad (12)$$

$\qquad\qquad \tau \leftarrow \mathcal{C}_\alpha \left(\dfrac{r}{n_2}\right)^{\beta/(2\beta + p \vee q)} \log(j)$;

$\qquad\qquad$ **if** *Val* $> \tau$ **then**

$\qquad\qquad\qquad$ ALARM $\leftarrow$ TRUE;

$\qquad\qquad\qquad$ **break**;

$\qquad$ **if** ALARM **then**

$\qquad\qquad$ **break**

---

In Theorem 1, we provide statistical guarantees for the false alarm probability and the detection delay of Algorithm 1.

**Theorem 1.** *Let the univariate orthonormal basis functions $\{\phi_k\}_{k \geq 1}$ in (2) be the Legendre polynomials. Suppose $N_{\text{train}}$, the size of the training data, is sufficiently large.*
*(a) Suppose the intensity of the data does not change, i.e.,*

$$\{X^{(i)}\}_{i=1}^{\infty} \overset{i.i.d.}{\sim} \lambda^*.$$

*Suppose $\|\lambda^*\|_{W^{\beta,2}} < \infty$, and the threshold constant $\mathcal{C}_\alpha$ in Algorithm 1 is chosen sufficiently large. Then, with probability at least $1 - \alpha$, Algorithm 1 never raises an alarm over the entire time horizon.*

*(b) Suppose the intensity changes at an unknown change point $\mathfrak{b}$ and*

$$\{X^{(i)}\}_{i=1}^{\mathfrak{b}} \overset{i.i.d.}{\sim} \lambda^* \quad \text{and} \quad \{X^{(i)}\}_{i=\mathfrak{b}+1}^{\infty} \overset{i.i.d.}{\sim} \lambda_a^*.$$

---

**Algorithm 2:** Restricted SVD

---

**Input**: Matrix $D \in \mathbb{R}^{M^p \times M^q}$; rank $r$; sample size $n_2$; dimensions $p, q$; smoothness $\beta > 0$

▶ **Adaptive trimming**;

$m \leftarrow \lceil (n_2/r)^{1/(2\beta + p \vee q)} \rceil$;

$\mathcal{B}_y \leftarrow \left\{ \phi_{i_1}(x_1) \cdots \phi_{i_p}(x_p) \right\}_{i_1,\ldots,i_p=1}^m, \quad \mathcal{B}_z \leftarrow \left\{ \phi_{\ell_1}(x_{p+1}) \cdots \phi_{\ell_q}(x_{p+q}) \right\}_{\ell_1,\ldots,\ell_q=1}^m$;

**for** $\mu \leftarrow 1$ **to** $M^p$ **do**
  **if** $\Phi_\mu \notin \mathcal{B}_y$ **then**
    set the $\mu$-th row $D_{\mu,*} \leftarrow 0$;

**for** $\eta \leftarrow 1$ **to** $M^q$ **do**
  **if** $\Psi_\eta \notin \mathcal{B}_z$ **then**
    set the $\eta$-th column $D_{*,\eta} \leftarrow 0$;

▶ **Rank-$r$ projection and score**;

$D[r] \leftarrow \mathrm{SVD}(D, r)$;

$\mathrm{Val} \leftarrow \|D[r]\|_{\mathrm{F}}$;

**Output**: Val

---

*Suppose $\|\lambda^*\|_{W^{\beta,2}}$ and $\|\lambda_a^*\|_{W^{\beta,2}}$ are both finite. Let $\{\sigma_k(\lambda^* - \lambda_a^*)\}_{k=1}^\infty$ be the singular values of $\lambda^* - \lambda_a^*$ as in (10), and suppose $r$ in Algorithm 1 is chosen so that*

$$\sqrt{\sum_{k=r+1}^\infty \sigma_k^2(\lambda^* - \lambda_a^*)} \leq \frac{\|\lambda^* - \lambda_a^*\|_{\mathbf{L}_2}}{5}. \tag{13}$$

*Let $\kappa = \|\lambda^* - \lambda_a^*\|_{\mathbf{L}_2}$ and*

$$\Delta = \left\lceil C_{\mathrm{lag}}\, r\, (\log(\mathfrak{b})/\kappa)^{2+(p \vee q)/\beta} \right\rceil, \tag{14}$$

*where $C_{\mathrm{lag}}$ is a sufficiently large constant depending only on $\mathcal{C}_\alpha$. Suppose in addition that the window size $W \geq \Delta$. Then, with probability at least $1 - \mathfrak{b}^{-3}$, Algorithm 1 raises an alarm within the time interval $(\mathfrak{b}, \mathfrak{b} + \Delta]$.*

Note that assumption (13) in Theorem 1 is a commonly condition in the literature. In particular, if the functional singular values $\{\sigma_k(\lambda^* - \lambda_a^*)\}_{k=1}^\infty$ decay at a polynomial or exponential rate, then (13) holds with a constant rank $r$; see Lemma 9. Such functional singular value decay assumptions are standard in the literature, see e.g., Hall et al. (2006); Raskutti et al. (2012).

Theorem 1 also implies that, when a change point is present, the detection delay is at most $O(\kappa^{-2-(p \vee q)/\beta})$. By comparison, Madrid Padilla et al. (2023) and Madrid Padilla et al. (2021) show that, for nonparametric density change point detection in the offline setting, the error scales as $O(\kappa^{-2-d/\beta})$, where $\kappa$ is the size of the change in $\mathbf{L}_2$-norm, $\beta$ is the degree of smoothness, and $d$ is the ambient dimension of the density function. Although we study Poisson intensity changes for streaming data, our detection delay bound is strictly better in order. This is because $p + q = d$, and thus $p \vee q < d$. So in the non-trivial setting that the change size $\kappa \ll 1$, we have

$$\kappa^{-2-(p \vee q)/\beta} \ll \kappa^{-2-d/\beta}.$$

## 3    SIMULATION STUDIES

We designed simulated experiments to evaluate the performance of Algorithm 1, which we refer to as the **Matrix detector**, against the following benchmarks. **Mean detector**: a simple vector detector that computes CUSUM statistics of observed vectors in a Poisson point process stream. **MMD detector**: following Li et al. (2015), we embed samples into a reproducing kernel Hilbert space and compute blockwise maximum mean discrepancy statistics between pre- and post-change windows. As a common choice in the literature, we use Gaussian kernel for the MMD detector. **KIE detector**: adapted from density change point detector from Madrid Padilla et al. (2023), we estimates the CUSUM statistics of intensity functions using kernel intensity estimator (KIE).

**Remark 4** (Choice of tuning parameters). *For the **MMD detector**, we set tuning parameters using the default procedure in Li et al. (2015); for the **KIE detector**, we follow the defaults in Madrid Padilla et al. (2023), i.e. we select the bandwidth of kernel using cross-validation in the training data. For our **Matrx detector**, we partition the coordinates into two groups so that, within the training data, variables are more correlated within groups and less correlated across groups. In addition, following common practice in the nonparametric literature (Wasserman, 2006), we set the smoothness parameter to $\beta = 2$. Given these choices, we select the remaining parameters $(r, W, C_\alpha)$ in Algorithm 1 by cross-validation on the training data. We also examine robustness of these choices in Figure 2.*

**The 3D setting.** In the first set of experiments, we generate Poisson point process data stream in $d = 3$ dimensions with the change point at time $\mathfrak{b} = 1200$. Before the change, each Poisson point process is drawn, using the thinning algorithm (Lewis & Shedler, 1979), from intensity

$$\lambda_1(x) = 5\{\sin(x_1 + x_2 + x_3) + 1\} + 5\{\cos(x_1 + x_2 + x_3) + 1\},$$

while after the change they are sampled from

$$\lambda_2(x) = a \cdot \exp\big(-(x_1 - 0.5)^2 - (x_2 - 0.5)^2 - (x_3 - 0.5)^2\big) \quad \text{with } a = 10 \text{ or } a = 20.$$

Note that $\|\lambda_1 - \lambda_2\|_{\mathbf{L}_2} \approx 7.23$ when $a = 10$, and $\|\lambda_1 - \lambda_2\|_{\mathbf{L}_2} \approx 2.94$ when $a = 20$. We refer to the case when $a = 10$ as the strong-signal case, and $a = 20$ as the weak-signal case. Each run consisted of $N_{\text{train}} = 1000$ pre-change samples and $N_{\text{total}} = 2000$ samples in total. All approaches were initialized on the same training data and applied to the same Monte Carlo replications (100 per threshold value). We evaluated 15 threshold values per method, ranging from loose (high sensitivity, high false alarm probability) to strict (low sensitivity, high delay). Performance was summarized using two metrics: (i) false alarm probability (FAP) and (ii) average detection delay (ADD). Note that if no alarm occurs by $N_{\text{total}}$, for comparison purpose, the delay of this replication is set to be $(N_{\text{total}} - \mathfrak{b})$.

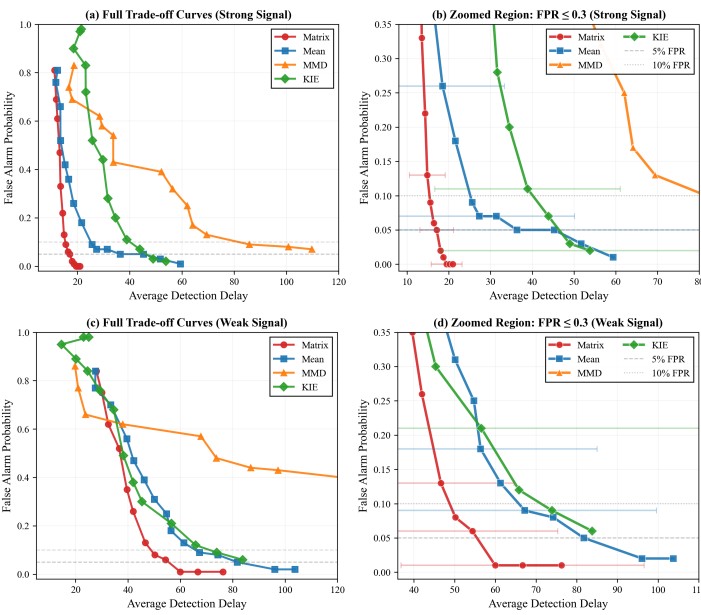

Figure 1: FAP vs ADD comparison among four detectors under strong-signal setting ($a = 10$, upper panels) and weak-signal setting ($a = 20$, lower panels).

Figure 1 presents the trade-off between empirical false alarm probability (FAP) and average detection delay (ADD). Our method (**Matrix**, red) consistently dominates the other three methods, achieving substantially lower detection delays for the same false alarm probability. Figure 1(b) and (d) summarize the performance of all methods when the FAP is controlled below 0.3 under both the strong-signal and weak-signal settings. Algorithm 1 achieves an ADD of about 17 (resp. 55) samples under strong-signal (resp. weak-signal) setting, compared to 36 (resp. 81) for the **Mean**

and 49 (resp. 85) for **KIE**. **MMD** did not attain an operating point below the 5% FAP threshold. Algorithm 1 consistently exhibits the smallest variability, indicating strong numerical stability.

We evaluate the robustness of Algorithm 1 to its hyper-parameters in the strong-signal setting $\lambda_2(x) = 10 \exp(-|x - 0.5|_2^2)$ of Section 3. We evaluate all feasible combinations of $(M, r)$ with $M \in \{2, 3, 4, 5\}$ and $r \in \{1, 2, 3, 4\}$ subject to $r \le M^{\min\{p,q\}}$ ($p + q = 3$). Detection thresholds are tuned automatically: we evaluate six candidate threshold factors and select the candidate with the minimum ADD subject to FAP $\le 0.5$; if no candidate attains FAP $\le 0.5$, we choose the one with the smallest FAP, breaking ties by smaller ADD.

Figure 2 summarizes the robustness study. Panel (a) shows that across all $(M, r)$ pairs, the **Matrix** detector maintains uniformly low FAP ($\le 5\%$), while ADD varies only mildly for moderate choices and increases noticeably only at the most over-parameterized settings (e.g., $M = 5, r \ge 3$; panel (b)). Panel (c) shows that runtime scales primarily with the basis size $M$ and is only weakly affected by $r$, indicating predictable computational cost. Panels (d)-(f) show that $M = 2$ and $r = 2$ maybe the best pair with controlled FAP, smallest ADD and short runtime.

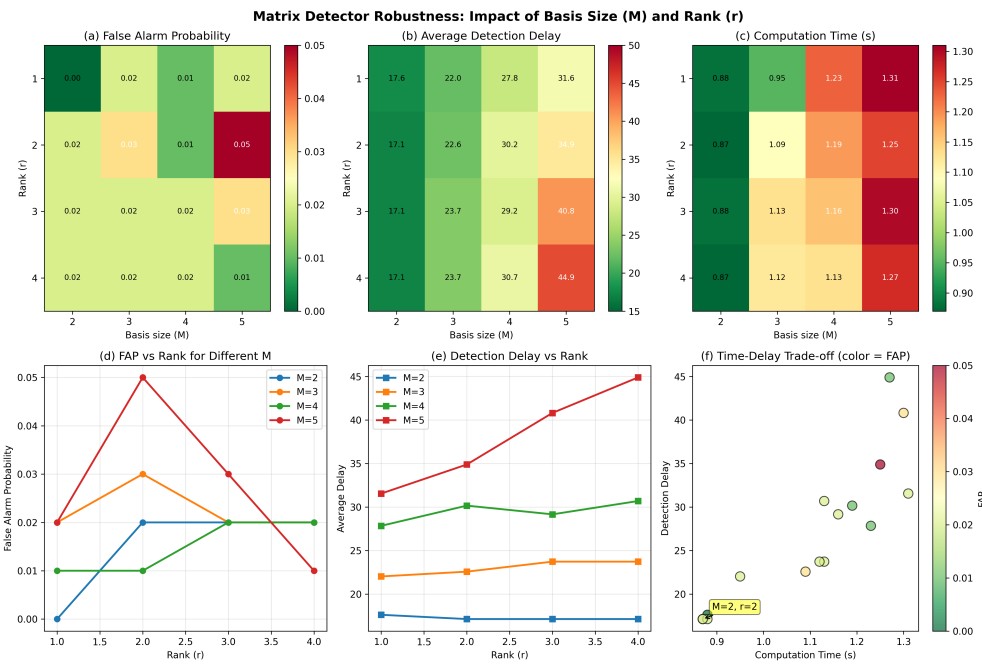

Figure 2: Robustness of the Matrix detector to basis size $M$ and target rank $r$ in terms of FAP, ADD and computational cost.

**The 4D setting.** We follow the setup in Section 3, except we consider Poisson point processes in dimension $d = 4$ with the pre-/post-change intensities

$$\lambda_3(x) = 2 \exp\left(-\frac{1}{8}\left\{\sum_{i=1}^{d-1} 0.01[(x_i - x_{i+1})(d + 1)]^2 + \sum_{i=1}^{d} 1.25\left(x_i^2 - 1\right)^2\right\}\right),$$

and

$$\lambda_4(x) = 5 \exp\left(-|x - 0.1|_2^2\right).$$

Figure 3 presents the trade-off between empirical FAP and average detection delay (ADD). Our method (**Matrix**, red) consistently dominates the baselines, achieving substantially smaller ADD at any fixed FAP. Panel (b) summarizes performance for operating points with FAP $\le 0.3$: the **Matrix** detector attains an ADD of about 33 samples, versus 54 for **KIE**; **Mean** and **MMD** do not reach FAP $\le 5\%$. We also display Monte Carlo standard errors as horizontal error bars; both **Matrix** and **KIE** exhibit small variability, indicating numerical stability.

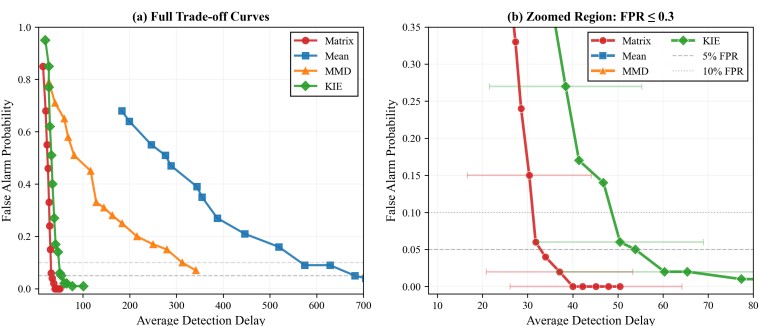

Figure 3: FAP vs ADD comparison among four detectors.

## 4 A Real Data Example

We study daily counts of new COVID-19 cases across all U.S. counties over a stream of 730 consecutive days spanning from the beginning of 2020 to the end of 2021. The data are obtained from the USAFacts repository.[1]

For each day, the dataset lists county locations and corresponding new-case counts. We treat each day's new cases in one day as one realization of a two-dimensional inhomogeneous Poisson process with coordinates given by the longitude and latitude of the corresponding county. To apply our modeling framework, we split the coordinates as $x =$ longitude and $y =$ latitude. In implementing Algorithm 1, we choose all the tuning parameters using cross-validation based on the training data, as discussed in Remark 4. The training period is September 1, 2020, to December 1, 2020.

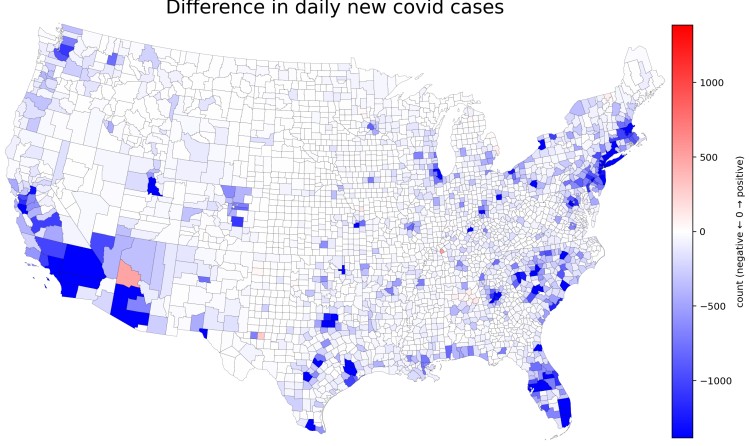

Figure 4: Difference in average daily new COVID-19 cases between the last two weeks of February and the first two weeks of March in 2021 at the county level. Blue denotes decreases; red denotes increases.

Our online procedure raised an alarm on March 2, 2021. This timing aligns with CDC reports[2] that indicate the new case counts peaked in late January 2021 and then declined rapidly through late February and early March 2021. The observed decline coincides with expanded public-health measures and the early effects of vaccine rollout. To further illustrate our findings, in Figure 4 we show the difference in average daily new cases between the last two weeks of February and the first two weeks of March 2021, where blue indicates decreases and red indicates increases.

[1]USAFacts COVID-19 confirmed cases
[2]CDC COVIDView, Feb. 26, 2021

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

# A  PROOF OF THEOREM 1

*Proof of Theorem 1.* Theorem 1 directly follows Lemma 1 and Lemma 2.  □

Let $\{X^{(i)}\}_{i=1}^{\infty}$ be a collection of independent Poisson processes where $X^{(i)} \subset \mathbb{R}^d$. Suppose $\{\phi_k\}_{k=1}^{\infty}$ is a collection of $\mathbf{L}_2(\mathbb{R})$ basis such that $\|\phi_k\|_\infty \leq C_\phi$ for some absolute constant $0 < C_\phi < \infty$. Let $p$ and $q$ be two positive integers such that

$$p + q = d \quad \text{and} \quad p \geq q.$$

For any $x = (x_1, \ldots, x_d) \in \mathbb{R}^d$, we can partition the coordinates as

$$x = (y, z), \quad \text{where} \quad y = (x_1, \ldots, x_p) \quad \text{and} \quad z = (x_{p+1}, \ldots, x_{p+q}).$$

Denote

$$\left\{ \phi_{\mu_1}(x_1) \cdots \phi_{\mu_p}(x_p) \right\}_{\mu_1, \ldots, \mu_p = 1}^{m} = \{\Phi_\mu(y)\}_{\mu=1}^{m^p} \quad \text{and}$$

$$\left\{ \phi_{\eta_1}(x_{p+1}) \cdots \phi_{\eta_q}(x_{p+q}) \right\}_{\eta_1, \ldots, \eta_q = 1}^{m} = \{\Psi_\eta(z)\}_{\eta=1}^{m^q}.$$

Let $\widehat{A}^{(i),m} \in \mathbb{R}^{m^p \times m^q}$ be such that the $\mu, \eta$ coordinate of $\widehat{A}^{(i),m}$ is

$$\widehat{A}^{(i),m}_{(\mu,\eta)} = \sum_{x=(y,z) \in X^{(i)}} \Phi_\mu(y)\Psi_\eta(z). \tag{15}$$

Given intervals $(0, t]$ and $[t, n]$, we define the CUSUM matrix $\widehat{A}^m_{n,t} \in \mathbb{R}^{m^p \times m^q}$ as

$$\widehat{A}^m_{n,t} = \frac{1}{t} \sum_{i=1}^{t} \widehat{A}^{(i),m} - \frac{1}{n-t} \sum_{i=t+1}^{n} \widehat{A}^{(i),m} \quad \text{and} \quad \widehat{A}^m_{n,t}[r] = \mathrm{SVD}(\widehat{A}^m_{n,t}, r). \tag{16}$$

In our approach, we choose the matrix size to be adaptive to $n$ and $t$. More precisely, we choose

$$m_{n,t} = \lceil \left(\frac{n-t}{r}\right)^{1/(2\beta+p)} \rceil,$$

and for brevity we denote

$$\widehat{A}_{n,t} = \widehat{A}^{m_{n,t}}_{n,t} \quad \text{and} \quad \widehat{A}_{n,t}[r] = \mathrm{SVD}(\widehat{A}_{n,t}, r). \tag{17}$$

Define the threshold parameter $\tau_{n,t}$ as

$$\tau_{n,t} = C_\alpha \left( \frac{r}{n-t} \right)^{\beta/(2\beta+p)} \log(n) \tag{18}$$

for some sufficiently large constant $C_\alpha > 0$ only depending on $\alpha$.

**Lemma 1.** *Suppose $m_{n,t} = \lceil \left(\frac{n-t}{r}\right)^{1/(2\beta+p)} \rceil$ and $\tau_{n,t}$ is defined in (18). Suppose $\{X^{(i)}\}_{i=1}^{\infty}$ is sampled from the same intensity function $\lambda^*$. Let $\widehat{A}_{n,t}[r]$ be defined as in (17). Then*

$$\mathbb{P}\left( \|\widehat{A}_{n,t}[r]\|_F \geq \tau_{n,t} \text{ for all } 1 \leq t < n < \infty \right) \leq 1 - \alpha.$$

*Proof.* Since there is no change points, we apply Lemma 4 to any give $t, n$ with the difference of ground truth intensity function $\delta^* = 0$ to have that with probability at most $\epsilon n^{-3}$,

$$\|\widehat{A}_{n,t}[r]\|_F \geq C_1 m_{n,t}^{-\beta} + C_\epsilon \sqrt{r}\{\log(m_{n,t}) + \log(n)\}\sqrt{\frac{m_{n,t}^p}{n-t}},$$

where we have used $\sigma_r(\delta^*) = 0$ when $\delta^* = 0$. Since $m_{n,t} = \lceil \left(\frac{n-t}{r}\right)^{1/(2\beta+p)} \rceil$, it follows that

$$\mathbb{P}\left( \|\widehat{A}_{n,t}[r]\|_F \geq C'_\epsilon \left(\frac{r}{n-t}\right)^{\beta/(2\beta+p)} \log(n) \right) \leq \epsilon n^{-3}.$$

By a union bound, for any $n \in \mathbb{Z}^+$ it holds that

$$\mathbb{P}\bigg(\|\widehat{A}_{n,t}[r]\|_{\mathrm{F}} \geq C'_\epsilon \big(\frac{r}{n-t}\big)^{\beta/(2\beta+p)} \log(n) \text{ for all } 1 \leq t < n\bigg) \leq \epsilon n^{-2}.$$

Since $\sum_{n=1}^\infty n^{-2} = \pi^2/6$, if follows that if $\epsilon = 6\alpha/\pi^2$, then by another union bound,

$$\mathbb{P}\bigg(\|\widehat{A}_{n,t}[r]\|_{\mathrm{F}} \geq C_\alpha \big(\frac{r}{n-t}\big)^{\beta/(2\beta+p)} \log(n) \text{ for all } 1 \leq t < n < \infty\bigg) \leq \alpha.$$

The desired result follows from the choice of $\tau_{n,t}$ in (18). $\qquad\square$

**Lemma 2.** *Let $\tau_{n,t}$ be defined as in (18). Suppose $\{X^{(i)}\}_{i=1}^{\mathfrak{b}}$ are sampled from the intensity function $\lambda^*$, and $\{X^{(i)}\}_{i=\mathfrak{b}+1}^\infty$ are sampled from the intensity function $\lambda_a^*$. Suppose that $p \geq q$ and that*

$$\sqrt{\sum_{k=r+1}^\infty \sigma_k^2(\lambda^* - \lambda_a^*)} \leq \frac{\|\lambda^* - \lambda_a^*\|_{\mathbf{L}_2}}{5}. \tag{19}$$

*Let $\kappa = \|\lambda^* - \lambda_a^*\|_{\mathbf{L}_2}$. Suppose in addition that*

$$\Delta \geq C_{\mathrm{lag}} r (\log(\mathfrak{b})/\kappa)^{2+p/\beta} \tag{20}$$

*where $C$ is a sufficiently large constant only depending on $C_\alpha$ in (18). Then*

$$\mathbb{P}\big(\|\widehat{A}_{\Delta+\mathfrak{b},\mathfrak{b}}[r]\|_{\mathrm{F}} \geq \tau_{\Delta+\mathfrak{b},\mathfrak{b}}\big) \geq 1 - \mathfrak{b}^{-3},$$

*where $\widehat{A}_{\Delta+\mathfrak{b},\mathfrak{b}}[r]$ is defined according to (17).*

*Proof.* We apply Lemma 4 to $\widehat{A}_{\Delta+\mathfrak{b},\mathfrak{b}}[r]$ with the difference of ground truth intensity function being $\delta^* = \lambda^* - \lambda_a^*$ to deduce that, with probability at least $1 - \epsilon\mathfrak{b}^{-3}$,

$$\left| \|\widehat{A}_{\Delta+\mathfrak{b},\mathfrak{b}}[r]\|_{\mathrm{F}} - \|\delta^*\|_{\mathbf{L}_2} \right|$$

$$\leq C_2 m_{\Delta+\mathfrak{b},\mathfrak{b}}^{-\beta} + C_\epsilon \sqrt{r} \big\{ \log(\mathfrak{b}) + \log(m_{\Delta+\mathfrak{b},\mathfrak{b}}) \big\} \sqrt{\frac{m_{\Delta+\mathfrak{b},\mathfrak{b}}^p}{\Delta}} + 4\sqrt{\sum_{k=r+1}^\infty \sigma_k^2(\delta^*)},$$

$$\leq C_2 m_{\Delta+\mathfrak{b},\mathfrak{b}}^{-\beta} + 2C_\epsilon \sqrt{r} \log(\mathfrak{b}) \sqrt{\frac{m_{\Delta+\mathfrak{b},\mathfrak{b}}^p}{\Delta}} + 4\sqrt{\sum_{k=r+1}^\infty \sigma_k^2(\delta^*)} \tag{21}$$

Here the second inequality follows from assumption that $N_{\mathrm{train}}$ is sufficiently large, $\mathfrak{b} \geq N_{\mathrm{train}}$, and the observation that

$$m_{\Delta+\mathfrak{b},\mathfrak{b}} = \lceil (\Delta/r)^{1/(2\beta+p)} \rceil \leq \mathfrak{b}.$$

It follows that with probability at least $1 - \epsilon\mathfrak{b}^{-3}$,

$$\|\widehat{A}_{\Delta+\mathfrak{b},\mathfrak{b}}[r]\|_{\mathrm{F}} \geq \frac{\|\delta^*\|_{\mathbf{L}_2}}{5} - \left( C_2 m_{\Delta+\mathfrak{b},\mathfrak{b}}^{-\beta} + 2C_\epsilon \sqrt{r} \log(\mathfrak{b}) \sqrt{\frac{m_{\Delta+\mathfrak{b},\mathfrak{b}}^p}{\Delta}} \right)$$

$$= \frac{\kappa}{5} - \left( C_2 m_{\Delta+\mathfrak{b},\mathfrak{b}}^{-\beta} + 2C_\epsilon \sqrt{r} \log(\mathfrak{b}) \sqrt{\frac{m_{\Delta+\mathfrak{b},\mathfrak{b}}^p}{\Delta}} \right)$$

$$\geq \frac{\kappa}{5} - C_3 \log(\mathfrak{b})(r/\Delta)^{\beta/(2\beta+p)} > \kappa/6, \tag{22}$$

where the first inequality follows from (21) and (19), the second inequality follows from $m_{\Delta+\mathfrak{b},\mathfrak{b}} = \lceil (\Delta/r)^{1/(2\beta+p)} \rceil$, and the last inequality follows from (20) with sufficiently large constant $C_{\mathrm{lag}}$. Note that

$$\tau_{\Delta+\mathfrak{b},\mathfrak{b}} = C_\alpha \left( \frac{r}{\Delta} \right)^{\beta/(2\beta+p)} \log(\Delta+\mathfrak{b}) \leq 2C_\alpha \left( \frac{r}{\Delta} \right)^{\beta/(2\beta+p)} \log(\mathfrak{b}) \leq \kappa/6. \tag{23}$$

Here the equality follows from (18), the first inequality follows from the fact that $\Delta \leq \mathfrak{b}$, and the second inequality follows from (20) with sufficiently large $C_{\mathrm{lag}}$. The desired result follows from (22) and (23). $\qquad\square$

# B    DEVIATION BOUNDS

Suppose $\{\phi_k\}_{k=1}^\infty$ is a collection of $\mathbf{L}_2(\mathbb{R})$ basis such that $\|\phi_k\|_\infty \leq C_\phi$ for some absolute constant $C_\phi$. Let $p$ and $q$ be two positive integers such that $p + q = d$. For any $x = (x_1, \ldots, x_d) \in \mathbb{R}^d$, we can partition the coordinates as

$$x = (y, z), \quad \text{where} \quad y = (x_1, \ldots, x_p) \quad \text{and} \quad z = (x_{p+1}, \ldots, x_{p+q}).$$

**Theorem 2.** *[Singular value decomposition in function space] Let $F(y, z) : \mathbb{R}^p \times \mathbb{R}^q \to \mathbb{R}$ be any function such that $\|F\|_{\mathbf{L}_2(\mathbb{R}^{p+q})} < \infty$. There exists a collection of singular values $\sigma_1(F) \geq \sigma_2(F) \geq \cdots \geq 0$, and two collections of orthonormal basis functions $\{f_\rho(y)\}_{\rho=1}^\infty \subset \mathbf{L}_2(\mathbb{R}^p)$ and $\{g_\rho(z)\}_{\rho=1}^\infty \subset \mathbf{L}_2(\mathbb{R}^q)$ such that*

$$F(y, z) = \sum_{\rho=1}^\infty \sigma_\rho(F) f_\rho(y) g_\rho(z). \tag{24}$$

*Proof.* See Section 6 of Brezis (2011). $\qquad\square$

Let $\delta^*(x) = \lambda^*(x) - \lambda_a^*(x) : \mathbb{R}^d \to \mathbb{R}$. Suppose the SVD of $\delta^*$ satisfies

$$\delta^*(x) = \delta^*(y, z) = \sum_{\rho=1}^\infty \sigma_\rho(\delta^*) f_\rho^*(y) g_\rho^*(z).$$

For a positive integer $r$, let

$$\delta^*[r](y, z) = \sum_{\rho=1}^r \sigma_\rho(\delta^*) f_\rho^*(y) g_\rho^*(z).$$

Therefore $\delta^*[r]$ is the best rank-$r$ estimate of $\delta^*$. We can represent $\delta^*$ as a matrix in the following way. Denote $y = (x_1, \ldots, x_p)$ and $z = (x_{p+1}, \ldots, x_{p+q})$. For a positive integer $m$, denote

$$\left\{ \phi_{\mu_1}(x_1) \cdots \phi_{\mu_p}(x_p) \right\}_{\mu_1, \ldots, \mu_p = 1}^m = \{\Phi_\mu(y)\}_{\mu=1}^{m^p} \quad \text{and} \tag{25}$$

$$\left\{ \phi_{\eta_1}(x_{p+1}) \cdots \phi_{\eta_q}(x_{p+q}) \right\}_{\eta_1, \ldots, \eta_q = 1}^m = \{\Psi_\eta(z)\}_{\eta=1}^{m^q}. \tag{26}$$

Let $A^* \in \mathbb{R}^{m^p \times m^q}$ be such that the $\mu, \eta$ entry of $A^*$ is

$$A_{(\mu, \eta)}^* = \iint_{\mathbb{R}^{p+q}} \delta^*(y, z) \Phi_\mu(y) \Psi_\eta(z) dy dz. \tag{27}$$

We can approximate $\delta^*$ using $A^*$ as follows. Let

$$\delta_m^*(y, z) = \sum_{\mu=1}^{m^p} \sum_{\eta=1}^{m^q} A_{(\mu, \eta)}^* \Phi_\mu(y) \Psi_\eta(z). \tag{28}$$

It was shown in Peng et al. (2024) appendix G1 that if $\{\phi\}_{k=1}^\infty$ are chosen to be the univariate Legendre polynomial basis function, then

$$\|\delta^* - \delta_m^*\|_{\mathbf{L}_2} \leq C \|\delta^*\|_{W^{2,\beta}} m^{-\beta} \tag{29}$$

where $\|\delta^*\|_{W^{2,\beta}}$ is the Sobolev norm of $\delta^*$.

**Definition 1.** *Let $N_1, N_2 \in \mathbb{Z}^+$ be such that $N_1 + N_2 \leq N$. Let $\{X^{(i)}\}_{i=1}^{N_1+N_2}$ be a collection of independent Poisson processes where $X^{(i)} \subset \mathbb{R}^d$. Suppose that the intensity function of $\{X^{(i)}\}_{i=1}^{N_1}$ is*

$$\lambda^*(x) : \mathbb{R}^d \to \mathbb{R}^+,$$

*and the intensity function of $\{X^{(i)}\}_{i=N_1+1}^{N_1+N_2}$ is*

$$\lambda_a^*(x) : \mathbb{R}^d \to \mathbb{R}^+.$$

Let $\widehat{A}^{(i)} \in \mathbb{R}^{m^p \times m^q}$ be such that

$$\widehat{A}^{(i)}_{(\mu,\eta)} = \sum_{x=(y,z)\in X^{(i)}} \Phi_\mu(y)\Psi_\eta(z).$$

Define

$$\widehat{A} = \frac{1}{N_1} \sum_{i=1}^{N_1} \widehat{A}^{(i)} - \frac{1}{N_2} \sum_{i=N_1+1}^{N_1+N_2} \widehat{A}^{(i)}. \tag{30}$$

With $x = (x_1, \ldots, x_d)$, $y = (x_1, \ldots, x_p)$ and $z = (x_{p+1}, \ldots, x_{p+q})$, we can write

$$\widehat{\delta}(y,z) = \sum_{\mu=1}^{m^p} \sum_{\eta=1}^{m^q} \widehat{A}_{(\mu,\eta)} \Phi_\mu(y) \Phi_\eta(z). \tag{31}$$

**Lemma 3.** *Let $\widehat{\delta}$ be defined as in* (31). *Then*

$$\mathbb{E}(\widehat{\delta}) = \delta_m^*.$$

*Proof.* Note that by Lemma 8, when $i \leq N_1$,

$$\mathbb{E}(\widehat{A}^{(i)}_{(\mu,\eta)}) = \iint_{\mathbb{R}^{p+q}} \Phi_\mu(y)\Psi_\eta(z)\lambda^*(y,z)dydz.$$

Therefore

$$\mathbb{E}(\widehat{A}_{(\mu,\eta)}) = \iint_{\mathbb{R}^{p+q}} \Phi_\mu(y)\Psi_\eta(z)(\lambda^*(y,z) - \lambda_a^*(y,z))dydz = A^*_{\mu,\eta}, \tag{32}$$

where the last equality follows from (27). Therefore

$$\mathbb{E}(\widehat{\delta}(y,z)) = \sum_{\mu=1}^{m^p} \sum_{\eta=1}^{m^q} \mathbb{E}(\widehat{A}_{(\mu,\eta)})\Phi_\mu(y)\Phi_\eta(z) = \sum_{\mu=1}^{m^p} \sum_{\eta=1}^{m^q} A^*_{(\mu,\eta)}\Phi_\mu(y)\Phi_\eta(z) = \delta_m^*(y,z).$$

$\square$

**Lemma 4.** *Let $\widehat{A} \in \mathbb{R}^{m^p \times m^q}$ be defined as in* (30) *and*

$$\widehat{A}[r] = \mathrm{SVD}(\widehat{A}, r).$$

*Suppose in that there exists an absolute constant $C_1$ such that*

$$\max\{\|\lambda^*\|_\infty, \|\lambda_a^*\|_\infty\} < C_1 \quad \max\{\|\lambda^*\|_{W^{2,\beta}}, \|\lambda_a^*\|_{W^{2,\beta}}\} < C_1,$$

*and that*

$$N_1 \geq N_2 \geq m^{\max\{p,q\}}. \tag{33}$$

*Then for any $\epsilon > 0$, with probability at most $\epsilon N^{-3}$, it holds that*

$$\left| \|\widehat{A}[r]\|_F - \|\delta^*\|_{L_2} \right| \geq C_2 m^{-\beta} + C_\epsilon \sqrt{r}\{\log(N) + \log(m)\}\sqrt{\frac{m^{\max\{p,q\}}}{N_2}} + 4\sqrt{\sum_{k=r+1}^\infty \sigma_k^2(\delta^*)},$$

*where $C_\epsilon$ is a positive constant only depending on $\epsilon$, and $C_2$ is a constant only depending on $C_1$.*

*Proof.* Let

$$\widehat{\delta}[r](y,z) = \sum_{\mu=1}^{m^p} \sum_{\eta=1}^{m^q} \widehat{A}[r]_{(\mu,\eta)}\Phi_\mu(y)\Phi_\eta(z).$$

Observe that

$$\|\delta^* - \widehat{\delta}[r]\|_{L_2} \leq \|\delta^* - \delta_m^*\|_{L_2} + \|\delta_m^* - \widehat{\delta}[r]\|_{L_2} = \|\delta^* - \delta_m^*\|_{L_2} + \|A^* - \widehat{A}[r]\|_F, \tag{34}$$

where $\delta_m^*$ is defined in (28), and the equality follows from Lemma 7. It follows from (29) that $\|\delta^* - \delta_m^*\|_{\mathbf{L}_2} \leq Cm^{-\beta}$. In addition, by Theorem 3,

$$\|A^* - \widehat{A}[r]\|_{\mathrm{F}} \leq 4\sqrt{\sum_{k=r+1}^{\infty} \sigma_k^2(A^*)} + 4\sqrt{r}\|A^* - \widehat{A}\|.$$

Note that

$$\sqrt{\sum_{k=r+1}^{\infty} \sigma_k^2(A^*)} = \sqrt{\sum_{k=r+1}^{\infty} \sigma_k^2(\delta_m^*)} \leq \sqrt{\sum_{k=r+1}^{\infty} \sigma_k^2(\delta_m^* - \delta^*)} + \sqrt{\sum_{k=r+1}^{\infty} \sigma_k^2(\delta^*)}$$

$$\leq \|\delta_m^* - \delta^*\|_{\mathrm{F}} + \sqrt{\sum_{k=r+1}^{\infty} \sigma_k^2(\delta^*)},$$

where the equality follows from Lemma 7, the first inequality follows from Lemma 6, and the second inequality follows from the fact that

$$\|\delta_m^* - \delta^*\|_{\mathrm{F}} = \sqrt{\sum_{k=1}^{\infty} \sigma_k^2(\delta_m^* - \delta^*)} \geq \sqrt{\sum_{k=r+1}^{\infty} \sigma_k^2(\delta_m^* - \delta^*)}.$$

Therefore

$$\|A^* - \widehat{A}[r]\|_{\mathrm{F}} \leq 4\|\delta_m^* - \delta^*\|_{\mathrm{F}} + 4\sqrt{\sum_{k=r+1}^{\infty} \sigma_k^2(\delta^*)} + 4\sqrt{r}\|A^* - \widehat{A}\|. \tag{35}$$

(34) and (35) together implie that

$$\|\delta^* - \widehat{\delta}[r]\|_{\mathbf{L}_2} \leq 5\|\delta_m^* - \delta^*\|_{\mathrm{F}} + 4\sqrt{\sum_{k=r+1}^{\infty} \sigma_k^2(\delta^*)} + 4\sqrt{r}\|A^* - \widehat{A}\|$$

$$\leq C_2 m^{-\beta} + 4\sqrt{\sum_{k=r+1}^{\infty} \sigma_k^2(\delta^*)} + 4\sqrt{r}\|A^* - \widehat{A}\|,$$

where the last inequality follows from (29). By Lemma 5 and the fact that $A^* = \mathbb{E}(\widehat{A})$ as in (32),

$$\mathbb{P}\bigg(\|\widehat{A} - A^*\| \geq t\bigg) \leq 2(m^p + m^q)\exp\bigg(\frac{-cN_2 t^2}{(m^p + m^q)C_1 + m^{d/2}t}\bigg).$$

It follows that there exists a constant $C_\epsilon$ only depending on $\epsilon$ such that

$$\mathbb{P}\bigg(\|\widehat{A} - A^*\| \geq C_\epsilon'\{\log(N) + \log(m)\}\bigg(\sqrt{\frac{m^p + m^q}{N_2}} + \frac{m^{d/2}}{N_2}\bigg)\bigg) \leq \epsilon N^{-3}.$$

Therefore with probability at least $1 - \epsilon N^{-3}$, it holds that

$$\left|\|\widehat{\delta}[r]\|_{\mathbf{L}_2} - \|\delta^*\|_{\mathbf{L}_2}\right| \leq \|\delta^* - \widehat{\delta}[r]\|_{\mathbf{L}_2}$$

$$\leq C_2 m^{-\beta} + C_\epsilon'\sqrt{r}\{\log(N) + \log(m)\}\bigg(\sqrt{\frac{m^p + m^q}{N_2}} + \frac{m^{d/2}}{N_2}\bigg) + 4\sqrt{\sum_{k=r+1}^{\infty} \sigma_k^2(\delta^*)}$$

$$\leq C_2 m^{-\beta} + C_\epsilon\sqrt{r}\{\log(N) + \log(m)\}\bigg(\sqrt{\frac{m^p + m^q}{N_2}}\bigg) + 4\sqrt{\sum_{k=r+1}^{\infty} \sigma_k^2(\delta^*)},$$

The last inequality follows from the fact that

$$\frac{m^{d/2}}{N_2} \leq \frac{m^{\max\{p,q\}}}{N_2} \leq \sqrt{\frac{m^{\max\{p,q\}}}{N_2}},$$

where the first inequality follows from $\max\{p, q\} \geq d/2$, then $p \geq d/2$, and the second inequality follows from (33). The desired result follows from the fact that $\|\widehat{\delta}[r]\|_{\mathbf{L}_2} = \|\widehat{A}[r]\|_{\mathrm{F}}$. □

### B.1 POISSON MATRIX PROPERTIES

**Lemma 5.** *Let $\widehat{A}$ be defined as in* (30)*. Then*

$$\mathbb{P}\left(\|\widehat{A} - \mathbb{E}(\widehat{A})\| \geq t\right) \leq 2(m^p + m^q)\exp\left(\frac{-cN_2 t^2}{(m^p + m^q)C_1 + m^{d/2}t}\right), \qquad (36)$$

*where $C_1 = \max\{\|\lambda^*\|_\infty, \|\lambda_a^*\|_\infty\}$.*

*Proof.* From Definition 1, $\{X^{(i)}\}_{i=1}^{N_1}$ are i.i.d. Poisson process. By Lemma 5 in Xu et al. (2025),

$$\mathbb{P}\left(\|\sum_{i=1}^{N_1}\widehat{A}^{(i)} - \mathbb{E}(\sum_{i=1}^{N_1}\widehat{A}^{(i)})\| \geq t\right) \leq (m^p + m^q)\exp\left(-\frac{ct^2}{(m^p + m^q)\|\lambda^*\|_\infty N_1 + m^{d/2}t}\right).$$

Similarly

$$\mathbb{P}\left(\|\sum_{i=N_1+1}^{N_1+N_2}\widehat{A}^{(i)} - \mathbb{E}(\sum_{i=N_1+1}^{N_1+N_2}\widehat{A}^{(i)})\| \geq t\right) \leq (m^p + m^q)\exp\left(-\frac{ct^2}{(m^p + m^q)\|\lambda^*\|_\infty N_2 + m^{d/2}t}\right).$$

The desired result follows from the fact that

$$\|\widehat{A} - \mathbb{E}(\widehat{A})\| \leq \frac{1}{N_1}\|\sum_{i=1}^{N_1}\widehat{A}^{(i)} - \mathbb{E}(\sum_{i=1}^{N_1}\widehat{A}^{(i)})\| + \frac{1}{N_2}\|\sum_{i=N_1+1}^{N_1+N_2}\widehat{A}^{(i)} - \mathbb{E}(\sum_{i=N_1+1}^{N_1+N_2}\widehat{A}^{(i)})\|.$$

$\square$

## C AUXILIARY RESULTS

**Lemma 6.** *[Mirsky in Hilbert space] Suppose $A$ and $B$ are two compact operators in $\mathcal{W} \otimes \mathcal{W}'$, where $\mathcal{W}$ and $\mathcal{W}'$ are two separable Hilbert spaces. Let $\{\sigma_k(\mathcal{A})\}_{k=1}^\infty$ be the singular values of $\mathcal{A}$ in the decreasing order, and $\{\sigma_k(\mathcal{B})\}_{k=1}^\infty$ be the singular values of $\mathcal{B}$ in the decreasing order. Then*

$$\sum_{k=1}^\infty (\sigma_k(A) - \sigma_k(B))^2 \leq \|A - B\|_F^2 = \sum_{k=1}^\infty \sigma_k^2(A - B).$$

*Proof.* This is Lemma 41 of Peng et al. (2024). $\square$

**Lemma 7.** *For any $A \in \mathbb{R}^{m^p \times m^q}$, let $\mathcal{F}$ be a map from $\mathbb{R}^{m^p \times m^q}$ to $\mathbf{L}_2(\mathbb{R}^p \times \mathbb{R}^p)$ such that*

$$\mathcal{F}(A)(y, z) = \sum_{\mu=1}^{m^p}\sum_{\eta=1}^{m^q} A_{\mu,\eta}\Phi_\mu(y)\Psi_\eta(z), \qquad (37)$$

*where $\{\Phi_\mu(y)\}_{\mu=1}^{m^p}$ and $\{\Psi_\mu(z)\}_{\eta=1}^{m^q}$ are generic basis functions of $\mathbf{L}_2(\mathbb{R}^p)$ and $\mathbf{L}_2(\mathbb{R}^q)$ respectively. Then*

$$\sigma_k(A) = \sigma_k(\mathcal{F}(A)) \quad \text{for any } k \in \mathbb{Z}^+.$$

*Proof.* Note that the map $F$ is distance preserving in the sense that for any $A, B \in \mathbb{R}^{m^p \times m^q}$,

$$\|A - B\|_F = \|\mathcal{F}(A) - \mathcal{F}(B)\|_{\mathbf{L}_2}.$$

Since $F$ is distance preserving, $F$ also preserves the singular values. $\square$

**Theorem 3.** *Let $X$ and $Z$ be two generic matrices in $\mathbb{R}^{p \times q}$ and that*

$$Y = X + Z.$$

*Denote $Y[r] = \mathrm{SVD}(Y, r)$. Then*

$$\|Y[r] - X^*\|_F \leq (2 + \sqrt{2})\left(\sqrt{\sum_{i=r+1}^{\min\{m,n\}}\sigma_i^2(X^*)} + \sqrt{r}\,\|Z\|\right), \qquad (38)$$

*Proof.* The desired result is a direct consequence of Lemma 16 Xu et al. (2025). □

**Lemma 8** (Campbell's Theorem). *Let $X \subset \mathbb{R}^d$ be a Poisson process with the intensity function $\lambda^*$. For all measurable function $f : \mathbb{R}^d \to \mathbb{R}$, it holds that*

$$\mathbb{E}(\sum_{x \in X} f(x)) = \int_{\mathbb{R}^d} f(x)\lambda^*(x)dx.$$

**Lemma 9.** *Suppose $\{\sigma_k(F)\}_{k=1}^\infty$ are the singular values the function $F$ and that $\{\sigma_k(F)\}_{k=1}^\infty$ decays at a polynomial rate. Then there exists a constant $r$ depending only on the decay rate of $\{\sigma_k(F)\}_{k=1}^\infty$ such that*

$$\sqrt{\sum_{k=r+1}^\infty \sigma_k^2(F)} \leq \frac{\|F\|_{\mathbf{L}_2}}{5}.$$

*Proof.* Without lost of generality, suppose that $\sigma_k(F) = ck^{-a}$ and $\|F\|_{\mathbf{L}_2} = 1$. Since

$$\sum_{k=r+1}^\infty \sigma_k^2(F) = \sum_{k=1}^\infty c^2 k^{-2a} = \|F\|_{\mathbf{L}_2}^2 = 1,$$

it follows that $a > 1/2$. Since $a > 1/2$,

$$\lim_{r \to \infty} \sum_{k=r+1}^\infty c^2 k^{-2a} = 0.$$

Therefore there exists $r \in \mathbb{Z}^+$ depending only on $a$ and $c$ such that $\sqrt{\sum_{k=r+1}^\infty \sigma_k^2(F)} \leq \frac{\|F\|_{\mathbf{L}_2}}{5}$
□

## D   ONLINE POISSON CHANGE POINT DETECTION IN 1D

For completeness, we introduce a simplified version of Algorithm 3 suitable for detecting change points for Poisson Process data stream in 1D.

Let $\{\phi_\mu\}_{\mu=1}^\infty$ be a collection of orthonormal basis functions of $\mathbf{L}_2(\mathbb{R})$, and $M$ be a postive integer. For each process $X^{(i)} \subset \mathbb{R}$, define its intensity matrix by

$$\widehat{\mathcal{V}}^{(i)} \in \mathbb{R}^M, \quad \widehat{\mathcal{V}}_\mu^{(i)} = \sum_{x \in X^{(i)}} \phi_\mu(x). \tag{39}$$

**Theorem 4.** *Let the univariate orthonormal basis functions $\{\phi_\mu\}_{\mu=1}^\infty$ in (39) be chosen as the Legendre polynomial basis functions, and $N_{\text{train}}$, the size of the training data be sufficiently large.*
*(a) Suppose intensity of the data does not change, i.e.,*

$$\{X^{(i)}\}_{i=1}^\infty \overset{i.i.d.}{\sim} \lambda^*.$$

*Suppose $\|\lambda^*\|_{W^{\beta,2}} < \infty$, and the threshold constant $\mathcal{C}_\alpha$ in Algorithm 3 is chosen sufficiently large. Then, with probability at least $1 - \alpha$, Algorithm 3 never raises an alarm over the entire time horizon.*

*(b) Suppose the intensity changes at an unknown change point $\mathfrak{b}$ and*

$$\{X^{(i)}\}_{i=1}^{\mathfrak{b}} \overset{i.i.d.}{\sim} \lambda^* \quad and \quad \{X^{(i)}\}_{i=\mathfrak{b}+1}^\infty \overset{i.i.d.}{\sim} \lambda_a^*.$$

*Suppose $\|\lambda^*\|_{W^{\beta,2}}$ and $\|\lambda_a^*\|_{W^{\beta,2}}$ are both finite. Let $\kappa = \|\lambda^* - \lambda_a^*\|_{\mathbf{L}_2}$ and*

$$\Delta = \left\lceil C_{\text{lag}} \left(\log(\mathfrak{b})/\kappa\right)^{2+1/\beta} \right\rceil, \tag{40}$$

*where $C_{\text{lag}}$ is a sufficiently large constant depending only on $\mathcal{C}_\alpha$. Suppose in addition that the window size $W \geq \Delta$. Then, with probability at least $1 - \mathfrak{b}^{-3}$, Algorithm 3 raises an alarm within the time interval $(\mathfrak{b}, \mathfrak{b} + \Delta]$.*

*Proof.* The proof of Theorem 4 is similar and simpler than Theorem 1, and will be omitted for brevity. □

---

**Algorithm 3:** Online Poisson change detection in 1D

---

**Input**: Smoothness parameter $\beta > 0$; window size $W$; threshold constant $\mathcal{C}_\alpha$

**Initialization Stage**;
▶ **Build a list $L$ of size $W$**;
**for** $k \leftarrow 1$ **to** $W$ **do**
$\qquad L[k] \leftarrow \sum_{i=1}^{N_{\text{train}}-W+k-1} \widehat{\mathcal{V}}^{(i)}$; here $\widehat{\mathcal{V}}^{(i)}$ is computed via (39) with $M = \left\lceil (W/r)^{1/(2\beta+1)} \right\rceil$

**Detection Stage**;
ALARM $\leftarrow$ FALSE;
**for** $j \leftarrow N_{\text{train}} + 1, N_{\text{train}} + 2, \ldots$ **do**
$\qquad$ ▶ **Update the list $L$**;
$\qquad$ **for** $k \leftarrow 1$ **to** $W - 1$ **do**
$\qquad \qquad L[k] \leftarrow L[k+1]$;
$\qquad L[W] \leftarrow L[W] + \widehat{\mathcal{V}}^{(j-1)}$;
$\qquad$ ▶ **Form the list $R$ of size $W$**;
$\qquad$ **for** $k \leftarrow 1$ **to** $W$ **do**
$\qquad \qquad R[k] \leftarrow \sum_{i=j-W+k}^{j} \widehat{\mathcal{V}}^{(i)}$;
$\qquad$ ▶ **Compute the CUSUM statistics in the sliding window**;
$\qquad$ **for** $k \leftarrow 1$ **to** $W$ **do**
$\qquad \qquad D \leftarrow \frac{L[k]}{n_1} - \frac{R[k]}{n_2} \in \mathbb{R}^M$ where $n_1 \leftarrow j - W - 1 + k, \quad n_2 \leftarrow W - k + 1$
$\qquad \qquad \|D\| \leftarrow \mathcal{C}_\alpha \left( \frac{r}{n_2} \right)^{\beta/(2\beta+p\vee q)} \log(j)$;
$\qquad \qquad$ **if** *Val* $> \tau$ **then**
$\qquad \qquad \qquad$ ALARM $\leftarrow$ TRUE;
$\qquad \qquad \qquad$ **break**;
$\qquad$ **if** ALARM **then**
$\qquad \qquad$ **break**

---

