# OpenReview forum: "Online Change Point Detection for Multivariate Poisson Point Processes"
_ICLR.cc/2026/Conference — Submitted to ICLR 2026_

### Official Review · Reviewer_DVQ3 · 2025-10-30

**Soundness:** 2
**Presentation:** 3
**Contribution:** 2
**Rating:** 2
**Confidence:** 4

**Summary:**

This paper addresses online change point detection for multivariate inhomogeneous Poisson point process (MIPPP) streams—a setting common in earthquake seismology, climate monitoring, and epidemic surveillance but underexplored in literature.

The core idea is to represent the MIPPP intensity function using low-rank matrices (via orthonormal basis expansion, e.g., Legendre polynomials) to enable nonparametric, adaptive change detection. The proposed algorithm processes data in a single pass with per-observation cost constant (independent of stream length), ensuring computational efficiency.

Key Results:
1.  A new online procedure for MIPPP change detection with linear total computational cost (scalable to long streams).

2.  Nonasymptotic bounds controlling (a) overall false alarm probability (≤α with high probability when no change occurs) and (b) detection delay (explicitly dependent on the \(L_2\)-norm jump \(\|\lambda^* - \lambda_a^*\|\) between pre- and post-change intensities).

3. (3D/4D MIPPP) and a real COVID-19 surveillance application (U.S. county-level cases) show the method outperforms baselines (Mean, MMD, KIE detectors) in reducing average detection delay (ADD) while maintaining low false alarm probability (FAP). For example, in strong-signal 3D settings, it achieves ADD≈17 (vs. 36 for Mean, 49 for KIE) with FAP≤0.3.

**Strengths:**

The paper demonstrates the originality by addressing a critical underexplored gap: online change point detection for multivariate inhomogeneous Poisson point process (MIPPP) streams.

The core innovation—using low-rank matrix representations (via orthonormal basis expansion, e.g., Legendre polynomials) to map continuous intensity functions to manageable matrices.

**Weaknesses:**

My main concern is the applicability and generalizability. See detailed comments in "Questions" below.

**Questions:**

1. The problem considered in the paper is oversimplified setting, where only at most one change point is assumed.
2. In the training phase, all data are assumed to follow a common intensity independently. I think this is a very strong assumption in the following sense.
2.a   Data cannot be viewed independently. There exists a temporal dependence between COVID cases from different days.
2.b   It is also not good to assume the  COVID processes on different days follow the same intensity. We all know that the COVID cases are not stable processes.
3. Apart from the real data (COVID-19) given in the paper, I wonder what other real data can be used in this paper? I feel like the number of real data can be applied here is limited.

4. Can the authors provide more motivations on why we split x in R^d into two parts, y in R^p, and z in R^q?

**Details Of Ethics Concerns:**

No concern.

---

> ### Author Response · Authors · 2025-11-21
> **Response to Reviewer DVQ3**
>
> We thank the reviewer for the thoughtful and constructive feedback. We will revise the manuscript substantially to clarify standard one change point simplification in online change point analysis, address temporal dependence in the application, and refine the real-data analysis. Below we respond point-by-point.
>
> **Question 1**
>
> Our focus on at most one change point is a standard simplification in online change point analysis. The detector is run in a detect–reset manner. Namely, once a change is flagged at $\hat{\eta}$, the algorithm is reinitialized on the post-$\hat{\eta}$ stream and continues scanning. This yields a practical multi-change procedure while allowing clear control of false alarms and local detection delay for each segment. However, this detect-reset manner requires the two consecutive true change points cannot be too close, i.e. less than the detection delay of a online change detector. We will add a remark clarifying this modeling choice and, for completeness, state the minimal-spacing condition under which repeated detect–reset yields valid performance guarantees over multiple changes.
>
> **Question 2**
>
> We agree that assuming independence across time and piecewise-constant intensities is too strong as stated. In the revision we will (i) replace the independence assumption with a short-range/mixing time-dependent framework and (ii) provide rigorous guarantees for the online detector under this dependence.  The (semi-)Markov Modulated Poisson Process serves as an example within the short-range/mixing time-dependent framework. In this example, the intensity remains constant across epochs but is influenced by a latent state with persistent dwell times. This model is commonly employed in the modeling of infectious disease dynamics.
>
> Relaxing the piecewise-constant intensity to a fully time-varying intensity would require redefining the target of inference, i.e. what constitutes a "change", which would shift the scope of this paper. We plan to pursue this in future work via a quasi-stationary segmentation perspective, where change points are defined as the boundaries of quasi-stationary segments (cf. the emerging literature on learning under non-stationarity). This allows controlled nonstationarity while preserving a clear and testable notion of change.
>
> **Question 3**
>
> Our method targets change points in a series of spatial (or marked) point processes, a data type that arises widely, not just in epidemiology. Typical examples include wildfires (Stoyan \& Penttinen, 2000; Waagepetersen, 2008; Møller \& Díaz-Avalos, 2010), earthquakes (Bray \& Schoenberg, 2013), city-wide crime incidents (Baddeley et al., 2021), and high-frequency financial transactions (Bauwens \& Hautsch, 2009). In all these settings, events arrive in space (and time), and monitoring proceeds epoch by epoch, exactly matching our streaming formulation. In particular, we plan to perform an additional real data analysis on earthquake activity in some states using United States Geological Survey (USGS) records from 2000 to 2025, aggregated weekly.
>
> **Question 4**
>
> The coordinate splitting is motivated from the following aspects.
>
> * **Modeling.**  In point process settings, coordinates are typically correlated. Such dependence creates natural groupings of variables (e.g. location vs. ancillary covariates). Modeling in these blocks lets us capture dominant between-block structure with a low-rank representation while allowing rich within-block correlations.
>
> * **Statistical efficiency.** The coordinate splitting unlock the low-rank approximation that can effective reduce the estimation variance.
>
> * **Computational efficiency.** The split reduces the representation from a $M^d$ tensor to an $M^p \times M^q$ matrix, enabling fast rank-$r$ SVD and light memory use. This makes online updates and scanning practical in real time.
>
> We will add a detailed remark on our motivations in the revision.

---

### Official Review · Reviewer_G654 · 2025-10-31

**Soundness:** 3
**Presentation:** 3
**Contribution:** 3
**Rating:** 6
**Confidence:** 3

**Summary:**

The paper studies the problem of detecting changes in an observed stream of Poisson point processes, in an online setting. To the best of their knowledge, there is no method that reliably detects changes in the intensity function (their main quantity of interest) in the change point literature. They represent the multivariate Poisson intensity function using low-rank matrices and propose an adaptive procedure to detect local changes in a nonparametric setting. They control the false alarm probability and quantify the detection delay both theoretically and experimentally using synthetic and real data.

**Strengths:**

The paper shows originality in that the proposed method involves both new techniques and yields favourable estimators in some setting. The technique of representing the intensity function using low-rank matrices is new, and deviates from other related work which is concerned with the more general problem of detecting changes in nonparametric densities (as far as I understand). Hence the innovation seems to come from specifying the distribution change detection problem to that of Poisson point processes, and using this distribution’s parametric structure to come up with linear algebraic techniques for the problem.

The presentation was reasonably clear.

The theoretical results seem sound, and the experimental (synthetic) results are convincing that this is the best method for estimating change points in the quantity of interest.

The work is significant. It experimentally improves over previous state of the art and admits a theoretical characterization which, once clarified, can make it a favourable method.

**Weaknesses:**

The innovation seems to come from specifying the problem to that of detecting the change in a specific distribution, and finding clever linear algebra representations that both produce a useful algorithm and a theoretical analysis. However, it is not surprising that designing a change point procedure for a distributional quantity that had not been studied before would yield better estimators.

Beyond the low rank matrix representation of the intensity function, the techniques used are standard in online change point analysis.

The real data example has no comparison, it would be nice to see how your method performs against others in this scenario.

**Questions:**

- Consequences of Theorem 1: you mention that your theoretical detection delay improves over that of Madrid Padilla in the “non-trivial” setting where \kappa << 1. Why is this setting non-trivial? What happens when \kappa is of order 1 or greater, and why is this not important? Are you implying that your method only does better in the case of vanishing signal? If so, do you have any intuition why the nonparametric method does better outside of this case?
- Before Remark 2, you state that the singular values of D in (9) decay at the same rate. Do you prove this? How important is this for the theoretical result/algorithm to work?
- Experimental work: the simulation studies are good and comparisons seem fair, and I liked that you compared in the setting where \kappa is of order 1, even though your theory does not indicate superiority in this regime (according to my understanding). Did you consider comparing against other models, such as the Madrid Padilla one, in the real data example? I think that would strengthen the paper, since from the theoretical result it is not clear how much of an improvement your method gives over Madrid Padilla.
- Typo at bottom of page 6: “We estimate the CUSUM statistics”.

---

> ### Author Response · Authors · 2025-11-21
> **Response to Reviewer G654**
>
> We thank the reviewer for the thoughtful and constructive feedback. We will revise the manuscript substantially to clarify different signal regimes, refine the numerical experiments, and correct typos. Below we respond point-by-point.
>
> **Questions 1**
>
> Thank you for raising this point. By "non-trivial" we mean the weak signal regime $\kappa \ll 1$ in the standard local-alternatives sense: detection is intrinsically hardest when the post-change signal is small relative to per-window noise. We do not intend to suggest that the moderate/strong-signal regimes ($\kappa = O(1)$ or $\kappa \gg 1$) are "trivial" or uninteresting. Rather, when the signal is moderate or strong, most reasonable procedures cross the threshold quickly and performance differences are dominated by operational factors (e.g. window length $W$) rather than statistical efficiency. The benefit of exploiting low-rank structure is most visible in the weak signal regime, while in moderate/strong signal regimes even nonparametric procedures perform comparably because the signal overwhelms noise.
>
> **Questions 2**
>
> In the revision, we will state explicitly the lemma below showing that the singular values of $\mathcal{M}(\lambda^\ast - \lambda_a^\ast)$ decay at the same rate as those of $(\lambda^* - \lambda_a^*)$. The proof is a direct consequence of the Eckart–Young–Mirsky theorem together with Mirsky’s inequality. This statement matters because it controls the approximation bias from projecting onto a finite tensor subspace. Our detection delay bounds then inherit the same spectral decay rates (though, for simplicity, we subsume this via assumption (13)). The algorithms themselves do not require to know this rate, but knowing it would help us to choose the rank $r$ under the alternatives.
>
> **Lemma:**
>
> Let $f\in L^2(\mathbb X_1\times\mathbb X_2)$ and let
> $\mathcal M f=\mathcal P_{\mathcal U}^{\top} \cdot f \cdot \mathcal P_{\mathcal V}$, where
> $\mathcal P_{\mathcal U}$ and $\mathcal P_{\mathcal V}$ are orthogonal projections onto
> finite-dimensional subspaces $\mathcal U\subset L^2(\mathbb X_1)$ and
> $\mathcal V\subset L^2(\mathbb X_2)$, respectively. Denote by
> $\sigma_k(g)$, $k \geq 1$, the singular values (in nonincreasing order) of the $g \in L^2(\mathbb X_1\times\mathbb X_2)$.
> Then for every integer $R\ge1$,
> $
> \sum_{k>R}\sigma_k^2(\mathcal M f) \le \sum_{k>R}\sigma_k^2(f).
> $
> Consequently, if $\sum_{k>R}\sigma_k^2(f) \le C R^{-2\alpha+1}$ (polynomial decay) or
> $\sum_{k>R}\sigma_k^2(f) \le C e^{-cR}$ (exponential decay) for some $C,c,\alpha>1$, then the
> same rate holds for $\mathcal M f$ with the same constants.
>
> **Questions 3**
>
> In our current experimental work (Section 3), we considered two regimes with normalized jump: a stronger-signal case $\kappa \approx 7.23$ and a weaker-signal case $\kappa \approx 2.94$. In the revision, we will expand the study to cover weak, moderate, and strong signals, i.e. $\kappa \ll 1$, $\kappa = O(1)$ and $\kappa \gg 1$. We will also add
>     the comparison against other competitors in the real data analysis.
>     We agree that these addition numerical experiments strengthen the paper.

---

### Official Review · Reviewer_d5pX · 2025-11-01

**Soundness:** 2
**Presentation:** 2
**Contribution:** 2
**Rating:** 2
**Confidence:** 3

**Summary:**

This paper studies online change-point detection for multivariate inhomogeneous Poisson point process data streams. The authors propose a low-rank representation of the multivariate Poisson intensity function, enabling an adaptive and nonparametric detection framework. The method is theoretically supported with guarantees on false alarm probability and detection delay.

**Strengths:**

- The paper addresses an underexplored problem of online intensity change detection under nonparametric and high-dimensional settings.
- Both new algorithms and theoretical guarantees are provided.

**Weaknesses:**

- The problem setup is vague and is not sufficiently justified. And some related prior work on Poisson process detection appears to be missing.
- The detection Algorithms depend on unspecified constants $C_\alpha$ and $C_{lag}$, making practical use difficult.
- The COVID-19 example is not very ideal to demonstrate the method’s strengths, and results are missing.
- The presentation could be significantly improved; the paper is difficult to follow in its current form.

**Questions:**

1. It should be made more clear in the problem setup what the meaning of streaming data is here. It seems the authors are assuming a series of point processes, not a stream of event data that is generated from a point process. If this is the case, then it should be mentioned how long each point process lasts, and does these point processes have any continuity in physical time. And it should be defined in the very beginning what is the format of each X^{(i)}. All these formulations need to be clarified to enhance the readability of the work.

2. The algorithm 1 and algorithm 2 have limited usage if the constants $C_\alpha$ and $C_{lag}$ are not specified: the detection threshold $\tau$ depends on $C_\alpha$, while the window size required depends on $C_{lag}$. Therefore, it must be specified what these constants are, rather than saying “sufficiently large” in order for the algorithm to be practically usable. It is mentioned in Remark 4 that “Given these choices, we select the remaining parameters (r,W,Cα) in Algorithm 1 by cross-validation on the training data.” It should be specified how this cross-validation is performed, and comment on what will be the case if there is no training data available for cross-validation.

3. The COVID dataset may not be an ideal real data example for this task, as the confirmed cases every day may have strong correlations, as there could be self-existing effects between confirmed cases, which makes it more meaningful to model the intensity function to be history dependent, such as a Hawkes model or an autoregressive model. Also, it is not clearly described how the authors convert the confirmed case data into Poisson processes. It seems to me that there should be a threshold such that the authors will mark an “event” in the location (long, lat) of the corresponding county if the confirmed cases in that county are above the threshold. And instead of presenting Figure 4, I think at least some kind of figures of the trajectory of the detection statistics should be presented to visualize the detection result; Figure 4 itself is not informative enough and is not helpful for demonstrating the effectiveness of the proposed method.

4. Some related prior work seems to be missing, such as the following. And I think there should be more related works on detecting changes in Poisson processes.
Nancy R. Zhang, Benjamin Yakir, Li C. Xia, and David Siegmund (2016). “Scan statistics on Poisson random fields with applications in genomics.” Ann. Appl. Stat., 10(2):726–755. https://doi.org/10.1214/15-AOAS892.

---

> ### Author Response · Authors · 2025-11-21
> **Response to Reviewer d5pX**
>
> We thank the reviewer for the thoughtful and constructive feedback. We will revise the manuscript substantially to clarify the problem setup, specify the tuning procedure, address temporal dependence in the application, and improve presentation. Below we respond point-by-point.
>
> **Questions 1 and 4**
>
> We agree and will make the setup explicit in our revision. We consider a series of spatial Poisson point processes observed at the end of fixed-length epochs $(t_{i - 1}, t_i]$, e.g. one day. Each $X^{(i)}$ is the finite set of event locations observed in epoch $i$. We will add a remark explaining that event-by-event streams can be handled by binning into epochs of length $\Delta$; changing $\Delta$ only rescales the per-epoch intensity and does not affect the methodology. In the COVID-19 example, $X^{(i)}$ is the set of (longitude, latitude) pairs for all new cases on day $i$.
>
> We will improve the presentation by providing more discussion on our theoretical results and adding guidance on the practical implementation of our algorithm. We will also revise the current numerical experiments and incorporate the suggested citations (e.g. Zhang et al., 2016) and additional related work on (Poisson) point-process change point detection.
>
> **Question 2**
>
> Algorithm 1's implementation indeed requires tuning parameters to be chosen, and we follow the hierarchy of these tuning parameters below. **(a) The rank $r$.** The low-rank projection in the sliding-window CUSUM balances bias and variance. When there is no change, the population CUSUM matrix is the zero matrix (rank $0$), so taking a small $r$ reduces variance without introducing systematic bias. In practice, the choice of $r$ is robust (see Fig. 2 in the paper). Thus, we set $r$ to a small constant, e.g.1–4.   **(b) Threshold $\tau$ and $C_{\alpha}$.** Both depend only on the pre-change distribution. As is standard in change-point analysis, we either assume the pre-change distribution is known or that a long stationary segment is available. Conditional on the chosen $r$, we calibrate $\tau$ as the $(1-\alpha)$-quantile of the null scan statistic computed on the reference segment (using permutation across epochs for i.i.d.data, or block permutation for short-range dependent data). The multiplicative constant $C_{\alpha}$ is thereby determined from the same reference data. **(c) $C_{lag}$.** This constant was only a proof artifact ensuring the window size $W$ satisfies a universal lower bound. We have removed $C_{lag}$ from the algorithms. The inputs now only require a user-specified $W$. **(d) The window size $W$.* $W$ should be large enough to control the estimation error of the jump signal. Given the sample size is $2000$, we set $W = 100$. We will add a detailed remark on the tuning parameter hierarchy and their practical choice in our revision.
>
> **Question 3**
>
> We acknowledge that the COVID-19 data may exhibit temporal dependence. In the revised version, we will adopt a general mixing time-dependent framework and provide a rigorous theoretical guarantee for the proposed online detection algorithm. The (semi-)Markov Modulated Poisson Process serves as an example within this framework. In this example, the intensity remains constant across epochs but is influenced by a latent state with persistent dwell times. This model is commonly employed in the modeling of infectious disease dynamics. Specifically, let $S_i\in\{1,\dots,M\}$ be a hidden semi–Markov chain with transition matrix $P$
> and state-specific dwell-time distributions $\{D_s\}$. Conditional on $S_i=s$,
> $X^{(i)}$ is a Poisson point process on $\mathcal D\subset\mathbb R^d$ with intensity
> $\lambda^{(s)}:\mathcal D\to(0,\infty)$ and likelihood
> $$
> \mathbb P\left(X^{(i)}| S_i=s\right)=
> \exp(-\Lambda^{(s)})\prod_{x\in X^{(i)}}\lambda^{(s)}(x),
> \quad
> \Lambda^{(s)}=\int_{\mathcal D}\lambda^{(s)}(x)dx.
> $$
> Hence the marginal intensity is piecewise constant in $i$:
> $\lambda_i(\cdot)=\lambda^{(S_i)}(\cdot)$; dependence across times arises through
> the persistence of $S_i$. Note that the i.i.d.~model
> (one state) is a special.

---

### Meta-Review · Area_Chair_aV7a · 2025-12-23

**Summary:**

The paper presents a novel approach for detecting changes in the intensity of streaming multivariate Poisson point processes.

Strengths

- The adaptation of low-rank matrix estimation techniques to functional intensity estimation for online change point detection is original.
- The proposed algorithm operates in a single pass with a computational cost per observation that is independent of the stream length.

Weaknesses
- Reviewers noted that the setup (treating sequential epochs as independent realizations of a Poisson process) is restrictive and often unrealistic for the targeted applications.
- The experimental section is a bit limited
- The theoretical guarantees rely on the independence of observations, which contradicts the nature of the real-world data used (COVID-19 cases).
- The practicality of the method is hindered by the need to tune multiple hyperparameters and the heuristic nature of the coordinate splitting strategy.

Overall, the paper is an interesting contribution but it is below the acceptance bar.

**Reviewer Concerns:**

The main concern that has not been addressed is about experimental quality and applicability of the method.

**Reviewer Scores:**

I think the first reviewer score may have increased but probably stayed under acceptance level

I think the second reviewers would have maintained the score

I think the third reviewer would have kept the score

---

### Decision · Program_Chairs · 2026-01-26

Reject